# Learning Imbalanced Datasets with Label-Distribution-Aware Margin Loss

**Kaidi Cao**
Stanford University
kaidicao@stanford.edu

**Colin Wei**
Stanford University
colinwei@stanford.edu

**Adrien Gaidon**
Toyota Research Institute
adrien.gaidon@tri.global

**Nikos Arechiga**
Toyota Research Institute
nikos.arechiga@tri.global

**Tengyu Ma**
Stanford University
tengyuma@stanford.edu

## Abstract

Deep learning algorithms can fare poorly when the training dataset suffers from heavy class-imbalance but the testing criterion requires good generalization on less frequent classes. We design two novel methods to improve performance in such scenarios. First, we propose a theoretically-principled label-distribution-aware margin (LDAM) loss motivated by minimizing a margin-based generalization bound. This loss replaces the standard cross-entropy objective during training and can be applied with prior strategies for training with class-imbalance such as re-weighting or re-sampling. Second, we propose a simple, yet effective, training schedule that defers re-weighting until after the initial stage, allowing the model to learn an initial representation while avoiding some of the complications associated with re-weighting or re-sampling. We test our methods on several benchmark vision tasks including the real-world imbalanced dataset iNaturalist 2018. Our experiments show that either of these methods alone can already improve over existing techniques and their combination achieves even better performance gains[1].

## 1 Introduction

Modern real-world large-scale datasets often have long-tailed label distributions [51, 28, 34, 12, 15, 50, 40]. On these datasets, deep neural networks have been found to perform poorly on less represented classes [17, 51, 5]. This is particularly detrimental if the testing criterion places more emphasis on minority classes. For example, accuracy on a uniform label distribution or the minimum accuracy among all classes are examples of such criteria. These are common scenarios in many applications [7, 42, 20] due to various practical concerns such as transferability to new domains, fairness, etc.

The two common approaches for learning long-tailed data are re-weighting the losses of the examples and re-sampling the examples in the SGD mini-batch (see [5, 21, 10, 17, 18, 9] and the references therein). They both devise a training loss that is in expectation closer to the test distribution, and therefore can achieve better trade-offs between the accuracies of the frequent classes and the minority classes. However, because we have fundamentally less information about the minority classes and the models deployed are often huge, over-fitting to the minority classes appears to be one of the challenges in improving these methods.

We propose to regularize the minority classes more strongly than the frequent classes so that we can improve the generalization error of minority classes without sacrificing the model's ability to fit

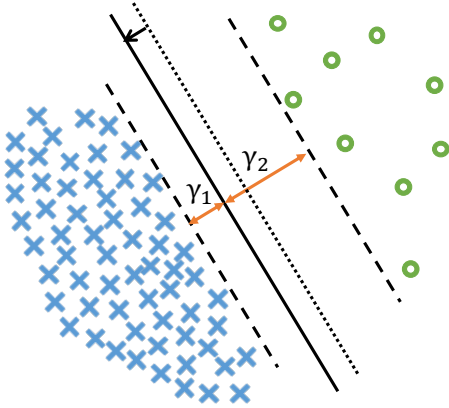

Figure 1: For binary classification with a linearly separable classifier, the margin $\gamma_i$ of the $i$-th class is defined to be the the minimum distance of the data in the $i$-th class to the decision boundary. We show that the test error with the uniform label distribution is bounded by a quantity that scales in $\frac{1}{\gamma_1\sqrt{n_1}} + \frac{1}{\gamma_2\sqrt{n_2}}$. As illustrated here, fixing the direction of the decision boundary leads to a fixed $\gamma_1 + \gamma_2$, but the trade-off between $\gamma_1, \gamma_2$ can be optimized by shifting the decision boundary. As derived in Section 3.1, the optimal trade-off is $\gamma_i \propto n_i^{-1/4}$ where $n_i$ is the sample size of the $i$-th class.

the frequent classes. Implementing this general idea requires a data-dependent or label-dependent regularizer — which in contrast to standard $\ell_2$ regularization depends not only on the weight matrices but also on the labels — to differentiate frequent and minority classes. The theoretical understanding of data-dependent regularizers is sparse (see [57, 43, 2] for a few recent works.)

We explore one of the simplest and most well-understood data-dependent properties: the margins of the training examples. Encouraging a large margin can be viewed as regularization, as standard generalization error bounds (e.g., [4, 59]) depend on the inverse of the minimum margin among all the examples. Motivated by the question of generalization with respect to minority classes, we instead study the minimum margin *per class* and obtain per-class and uniform-label test error bounds.[2] Minimizing the obtained bounds gives an optimal trade-off between the margins of the classes. See Figure 1 for an illustration in the binary classification case.

Inspired by the theory, we design a label-distribution-aware loss function that encourages the model to have the optimal trade-off between per-class margins. The proposed loss extends the existing soft margin loss [53] by encouraging the minority classes to have larger margins. As a label-dependent regularization technique, our modified loss function is orthogonal to the re-weighting and re-sampling approach. In fact, we also design a deferred re-balancing optimization procedure that allows us to combine the re-weighting strategy with our loss (or other losses) in a more efficient way.

In summary, our main contributions are (i) we design a label-distribution-aware loss function to encourage larger margins for minority classes, (ii) we propose a simple deferred re-balancing optimization procedure to apply re-weighting more effectively, and (iii) our practical implementation shows significant improvements on several benchmark vision tasks, such as artificially imbalanced CIFAR and Tiny ImageNet [1], and the real-world large-scale imbalanced dataset iNaturalist'18 [52].

## 2   Related Works

Most existing algorithms for learning imbalanced datasets can be divided in to two categories: re-sampling and re-weighting.

**Re-sampling.** There are two types of re-sampling techniques: over-sampling the minority classes (see e.g., [46, 60, 5, 6] and references therein) and under-sampling the frequent classes (see, e.g., [17, 23, 5] and the references therein.) The downside of under-sampling is that it discards a large portion of the data and thus is not feasible when data imbalance is extreme. Over-sampling is effective in a lot of cases but can lead to over-fitting of the minority classes [9, 10]. Stronger data augmentation for minority classes can help alleviate the over-fitting [9, 61].

**Re-weighting.** Cost-sensitive re-weighting assigns (adaptive) weights for different classes or even different samples. The vanilla scheme re-weights classes proportionally to the inverse of their frequency [21, 22, 55]. Re-weighting methods tend to make the optimization of deep models difficult under extreme data imbalanced settings and large-scale scenarios [21, 22]. Cui et al. [10] observe

that re-weighting by inverse class frequency yields poor performance on frequent classes, and thus propose re-weighting by the inverse effective number of samples. This is the main prior work that we empirically compare with.

Another line of work assigns weights to each sample based on their individual properties. Focal loss [35] down-weights the well-classified examples; Li et al. [31] suggests an improved technique which down-weights examples with either very small gradients or large gradients because examples with small gradients are well-classified and those with large gradients tend to be outliers.

In a recent work [6], Byrd and Lipton study the effect of importance weighting and show that empirically importance weighting does not have a significant effect when no regularization is applied, which is consistent with the theoretical prediction in [48] that logistical regression without regularization converges to the max margin solution. In our work, we explicitly encourage rare classes to have higher margin, and therefore we don't converge to a max margin solution. Moreover, in our experiments, we apply non-trivial $\ell_2$-regularization to achieve the best generalization performance. We also found deferred re-weighting (or deferred re-sampling) are more effective than re-weighting and re-sampling from the beginning of the training.

In contrast, and orthogonally to these papers above, our main technique aims to improve the generalization of the minority classes by applying additional regularization that is orthogonal to the re-weighting scheme. We also propose a deferred re-balancing optimization procedure to improve the optimization and generalization of a generic re-weighting scheme.

**Margin loss.** The hinge loss is often used to obtain a "max-margin" classifier, most notably in SVMs [49]. Recently, Large-Margin Softmax [37], Angular Softmax [38], and Additive Margin Softmax [53] have been proposed to minimize intra-class variation in predictions and enlarge the inter-class margin by incorporating the idea of angular margin. In contrast to the class-independent margins in these papers, our approach encourages bigger margins for minority classes. Uneven margins for imbalanced datasets are also proposed and studied in [32] and the recent work [25, 33]. Our theory put this idea on a more theoretical footing by providing a concrete formula for the desired margins of the classes alongside good empirical progress.

**Label shift in domain adaptation.** The problem of learning imbalanced datasets can be also viewed as a label shift problem in transfer learning or domain adaptation (for which we refer the readers to the survey [54] and the reference therein). In a typical label shift formulation, the difficulty is to detect and estimate the label shift, and after estimating the label shift, re-weighting or re-sampling is applied. We are addressing a largely different question: can we do better than re-weighting or re-sampling when the label shift is known? In fact, our algorithms can be used to replace the re-weighting steps of some of the recent interesting work on detecting and correcting label shift [36, 3].

Distributionally robust optimization (DRO) is another technique for domain adaptation (see [11, 16, 8] and the reference therein.) However, the formulation assumes no knowledge of the target label distribution beyond a bound on the amount of shift, which makes the problem very challenging. We here assume the knowledge of the test label distribution, using which we design efficient methods that can scale easily to large-scale vision datasets with significant improvements.

**Meta-learning.** Meta-learning is also used in improving the performance on imbalanced datasets or the few shot learning settings. We refer the readers to [55, 47, 56] and the references therein. So far, we generally believe that our approaches that modify the losses are more computationally efficient than meta-learning based approaches.

# 3 Main Approach

## 3.1 Theoretical Motivations

**Problem setup and notations.** We assume the input space is $\mathbb{R}^d$ and the label space is $\{1, \ldots, k\}$. Let $x$ denote the input and $y$ denote the corresponding label. We assume that the class-conditional distribution $\mathcal{P}(x \mid y)$ is the same at training and test time. Let $\mathcal{P}_j$ denote the class-conditional distribution, i.e. $\mathcal{P}_j = \mathcal{P}(x \mid y = j)$. We will use $\mathcal{P}_{\text{bal}}$ to denote the balanced test distribution which first samples a class uniformly and then samples data from $\mathcal{P}_j$.

For a model $f : \mathbb{R}^d \to \mathbb{R}^k$ that outputs $k$ logits, we use $L_{\text{bal}}[f]$ to denote the standard 0-1 test error on the balanced data distribution:

$$L_{\text{bal}}[f] = \Pr_{(x,y) \sim \mathcal{P}_{\text{bal}}} [f(x)_y < \max_{\ell \neq y} f(x)_\ell]$$

Similarly, the error $L_j$ for class $j$ is defined as $L_j[f] = \Pr_{(x,y) \sim \mathcal{P}_j} [f(x)_y < \max_{\ell \neq y} f(x)_\ell]$. Suppose we have a training dataset $\{(x_i, y_i)\}_{i=1}^n$. Let $n_j$ be the number of examples in class $j$. Let $S_j = \{i : y_i = j\}$ denote the example indices corresponding to class $j$.

Define the margin of an example $(x, y)$ as

$$\gamma(x, y) = f(x)_y - \max_{j \neq y} f(x)_j \tag{1}$$

Define the training margin for class $j$ as:

$$\gamma_j = \min_{i \in S_j} \gamma(x_i, y_i) \tag{2}$$

We consider the separable cases (meaning that all the training examples are classified correctly) because neural networks are often over-parameterized and can fit the training data well. We also note that the minimum margin of all the classes, $\gamma_{\min} = \min\{\gamma_1, \ldots, \gamma_k\}$, is the classical notion of training margin studied in the past [27].

**Fine-grained generalization error bounds.** Let $\mathcal{F}$ be the family of hypothesis class. Let $\mathrm{C}(\mathcal{F})$ be some proper complexity measure of the hypothesis class $\mathcal{F}$. There is a large body of recent work on measuring the complexity of neural networks (see [4, 13, 57] and references therein), and our discussion below is orthogonal to the precise choices. When the training distribution and the test distribution are the same, the typical generalization error bounds scale in $\mathrm{C}(\mathcal{F})/\sqrt{n}$. That is, in our case, if the test distribution is also imbalanced as the training distribution, then

$$\text{imbalanced test error} \lesssim \frac{1}{\gamma_{\min}} \sqrt{\frac{\mathrm{C}(\mathcal{F})}{n}} \tag{3}$$

Note that the bound is oblivious to the label distribution, and only involves the minimum margin across all examples and the total number of data points. We extend such bounds to the setting with balanced test distribution by considering the margin of each class. As we will see, the more fine-grained bound below allows us to design new training loss function that is customized to the imbalanced dataset.

**Theorem 1** (Informal and simplified version of Theorem 2). *With high probability $(1 - n^{-5})$ over the randomness of the training data, the error $L_j$ for class $j$ is bounded by*

$$L_j[f] \lesssim \frac{1}{\gamma_j} \sqrt{\frac{\mathrm{C}(\mathcal{F})}{n_j}} + \frac{\log n}{\sqrt{n_j}} \tag{4}$$

*where we use $\lesssim$ to hide constant factors. As a direct consequence,*

$$L_{\text{bal}}[f] \lesssim \frac{1}{k} \sum_{j=1}^{k} \left( \frac{1}{\gamma_j} \sqrt{\frac{\mathrm{C}(\mathcal{F})}{n_j}} + \frac{\log n}{\sqrt{n_j}} \right) \tag{5}$$

**Class-distribution-aware margin trade-off.** The generalization error bound (4) for each class suggests that if we wish to improve the generalization of minority classes (those with small $n_j$'s), we should aim to enforce bigger margins $\gamma_j$'s for them. However, enforcing bigger margins for minority classes may hurt the margins of the frequent classes. What is the optimal trade-off between the margins of the classes? An answer for the general case may be difficult, but fortunately we can obtain the optimal trade-off for the binary classification problem.

With $k = 2$ classes, we aim to optimize the balanced generalization error bound provided in (5), which can be simplified to (by removing the low order term $\frac{\log n}{\sqrt{n_j}}$ and the common factor $\mathrm{C}(\mathcal{F})$)

$$\frac{1}{\gamma_1 \sqrt{n_1}} + \frac{1}{\gamma_2 \sqrt{n_2}} \tag{6}$$

At the first sight, because $\gamma_1$ and $\gamma_2$ are complicated functions of the weight matrices, it appears difficult to understand the optimal margins. However, we can figure out the relative scales between $\gamma_1$ and $\gamma_2$. Suppose $\gamma_1, \gamma_2 > 0$ minimize the equation above, we observe that any $\gamma_1' = \gamma_1 - \delta$ and $\gamma_2' = \gamma_2 + \delta$ (for $\delta \in (-\gamma_2, \gamma_1)$) can be realized by the same weight matrices with a shifted bias term (See Figure 1 for an illustration). Therefore, for $\gamma_1, \gamma_2$ to be optimal, they should satisfy

$$\frac{1}{\gamma_1\sqrt{n_1}} + \frac{1}{\gamma_2\sqrt{n_2}} \leq \frac{1}{(\gamma_1 - \delta)\sqrt{n_1}} + \frac{1}{(\gamma_2 + \delta)\sqrt{n_2}} \tag{7}$$

The equation above implies that

$$\gamma_1 = \frac{C}{n_1^{1/4}}, \text{ and } \gamma_2 = \frac{C}{n_2^{1/4}} \tag{8}$$

for some constant $C$. Please see a detailed derivation in the Section A.

**Fast rate vs slow rate, and the implication on the choice of margins.** The bound in Theorem 1 may not necessarily be tight. The generalization bounds that scale in $1/\sqrt{n}$ (or $1/\sqrt{n_i}$ here with imbalanced classes) are generally referred to the "slow rate" and those that scale in $1/n$ are referred to the "fast rate". With deep neural networks and when the model is sufficiently big enough, it is possible that some of these bounds can be improved to the fast rate. See [58] for some recent development. In those cases, we can derive the optimal trade-off of the margin to be $n_i \propto n_i^{-1/3}$.

### 3.2 Label-Distribution-Aware Margin Loss

Inspired by the trade-off between the class margins in Section 3.1 for two classes, we propose to enforce a class-dependent margin for multiple classes of the form

$$\gamma_j = \frac{C}{n_j^{1/4}} \tag{9}$$

We will design a soft margin loss function to encourage the network to have the margins above. Let $(x, y)$ be an example and $f$ be a model. For simplicity, we use $z_j = f(x)_j$ to denote the $j$-th output of the model for the $j$-th class.

The most natural choice would be a multi-class extension of the hinge loss:

$$\mathcal{L}_{\text{LDAM-HG}}((x,y); f) = \max(\max_{j \neq y}\{z_j\} - z_y + \Delta_y, 0) \tag{10}$$

$$\text{where } \Delta_j = \frac{C}{n_j^{1/4}} \text{ for } j \in \{1, \ldots, k\} \tag{11}$$

Here $C$ is a hyper-parameter to be tuned. In order to tune the margin more easily, we effectively normalize the logits (the input to the loss function) by normalizing last hidden activation to $\ell_2$ norm 1, and normalizing the weight vectors of the last fully-connected layer to $\ell_2$ norm 1, following the previous work [53]. Notice that we then scale the logits by a constant $s = 10$ following [53]. Empirically, the non-smoothness of hinge loss may pose difficulties for optimization. The smooth relaxation of the hinge loss is the following cross-entropy loss with enforced margins:

$$\mathcal{L}_{\text{LDAM}}((x,y); f) = -\log \frac{e^{z_y - \Delta_y}}{e^{z_y - \Delta_y} + \sum_{j \neq y} e^{z_j}} \tag{12}$$

$$\text{where } \Delta_j = \frac{C}{n_j^{1/4}} \text{ for } j \in \{1, \ldots, k\} \tag{13}$$

In the previous work [37, 38, 53] where the training set is usually balanced, the margin $\Delta_y$ is chosen to be a label independent constant $C$, whereas our margin depends on the label distribution.

*Remark:* Attentive readers may find the loss $\mathcal{L}_{\text{LDAM}}$ somewhat reminiscent of the re-weighting because in the binary classification case — where the model outputs a single real number which is

passed through a sigmoid to be converted into a probability, — both the two approaches change the gradient of an example by a scalar factor. However, we remark two key differences: the scalar factor introduced by the re-weighting only depends on the class, whereas the scalar introduced by $\mathcal{L}_{\text{LDAM}}$ also depends on the output of the model; for multiclass classification problems, the proposed loss $\mathcal{L}_{\text{LDAM}}$ affects the gradient of the example in a more involved way than only introducing a scalar factor. Moreover, recent work has shown that, under separable assumptions, the logistical loss, with weak regularization [59] or without regularization [48], gives the max margin solution, which is in turn not effected by any re-weighting by its definition. This further suggests that the loss $\mathcal{L}_{\text{LDAM}}$ and the re-weighting may complement each other, as we have seen in the experiments. (Re-weighting would affect the margin in the non-separable data case, which is left for future work.)

### 3.3 Deferred Re-balancing Optimization Schedule

Cost-sensitive re-weighting and re-sampling are two well-known and successful strategies to cope with imbalanced datasets because, in expectation, they effectively make the imbalanced training distribution closer to the uniform test distribution. The known issues with applying these techniques are (a) re-sampling the examples in minority classes often causes heavy over-fitting to the minority classes when the model is a deep neural network, as pointed out in prior work (e.g., [10]), and (b) weighting up the minority classes' losses can cause difficulties and instability in optimization, especially when the classes are extremely imbalanced [10, 21]. In fact, Cui et al. [10] develop a novel and sophisticated learning rate schedule to cope with the optimization difficulty.

We observe empirically that re-weighting and re-sampling are both inferior to the vanilla empirical risk minimization (ERM) algorithm (where all training examples have the same weight) before annealing the learning rate in the following sense. The features produced before annealing the learning rate by re-weighting and re-sampling are worse than those produced by ERM. (See Figure 6 for an ablation study of the feature quality performed by training linear classifiers on top of the features on a large balanced dataset.)

Inspired by this, we develop a deferred re-balancing training procedure (Algorithm 1), which first trains using vanilla ERM with the LDAM loss before annealing the learning rate, and then deploys a re-weighted LDAM loss with a smaller learning rate. Empirically, the first stage of training leads to a good initialization for the second stage of training with re-weighted losses. Because the loss is non-convex and the learning rate in the second stage is relatively small, the second stage does not move the weights very far. Interestingly, with our LDAM loss and deferred re-balancing training, the vanilla re-weighting scheme (which re-weights by the inverse of the number of examples in each class) works as well as the re-weighting scheme introduced in prior work [10]. We also found that with our re-weighting scheme and LDAM, we are less sensitive to early stopping than [10].

---
**Algorithm 1** Deferred Re-balancing Optimization with LDAM Loss
---
**Require:** Dataset $\mathcal{D} = \{(x_i, y_i)\}_{i=1}^{n}$. A parameterized model $f_\theta$
 1: Initialize the model parameters $\theta$ randomly
 2: **for** $t = 1$ to $T_0$ **do**
 3:      $\mathcal{B} \leftarrow \text{SampleMiniBatch}(\mathcal{D}, m)$                             ▷ a mini-batch of $m$ examples
 4:      $\mathcal{L}(f_\theta) \leftarrow \frac{1}{m} \sum_{(x,y) \in \mathcal{B}} \mathcal{L}_{\text{LDAM}}((x, y); f_\theta)$
 5:      $f_\theta \leftarrow f_\theta - \alpha \nabla_\theta \mathcal{L}(f_\theta)$                                     ▷ one SGD step
 6:      Optional: $\alpha \leftarrow \alpha / \tau$                    ▷ anneal learning rate by a factor $\tau$ if necessary
 7:
 8: **for** $t = T_0$ to $T$ **do**
 9:      $\mathcal{B} \leftarrow \text{SampleMiniBatch}(\mathcal{D}, m)$                           ▷ A mini-batch of $m$ examples
10:     $\mathcal{L}(f_\theta) \leftarrow \frac{1}{m} \sum_{(x,y) \in \mathcal{B}} n_y^{-1} \cdot \mathcal{L}_{\text{LDAM}}((x, y); f_\theta)$      ▷ standard re-weighting by frequency
11:     $f_\theta \leftarrow f_\theta - \alpha \frac{1}{\sum_{(x,y) \in \mathcal{B}} n_y^{-1}} \nabla_\theta \mathcal{L}(f_\theta)$      ▷ one SGD step with re-normalized learning rate
---

## 4 Experiments

We evaluate our proposed algorithm on artificially created versions of IMDB review [41], CIFAR-10, CIFAR-100 [29] and Tiny ImageNet [45, 1] with controllable degrees of data imbalance, as well as a

Table 1: Top-1 validation errors on imbalanced IMDB review dataset. Our proposed approach LDAM-DRW outperforms the baselines.

| Approach | Error on positive reviews | Error on negative reviews | Mean Error |
|----------|---------------------------|---------------------------|------------|
| ERM | 2.86 | 70.78 | 36.82 |
| RS | 7.12 | 45.88 | 26.50 |
| RW | 5.20 | 42.12 | 23.66 |
| LDAM-DRW | 4.91 | 30.77 | 17.84 |

real-world large-scale imbalanced dataset, iNaturalist 2018 [52]. Our core algorithm is developed using PyTorch [44].

**Baselines.** We compare our methods with the standard training and several state-of-the-art techniques and their combinations that have been widely adopted to mitigate the issues with training on imbalanced datasets: (1) Empirical risk minimization (ERM) loss: all the examples have the same weights; by default, we use standard cross-entropy loss. (2) Re-Weighting (RW): we re-weight each sample by the inverse of the sample size of its class, and then re-normalize to make the weights 1 on average in the mini-batch. (3) Re-Sampling (RS): each example is sampled with probability proportional to the inverse sample size of its class. (4) CB [10]: the examples are re-weighted or re-sampled according to the inverse of the effective number of samples in each class, defined as $(1 - \beta^{n_i})/(1 - \beta)$, instead of inverse class frequencies. This idea can be combined with either re-weighting or re-sampling. (5) Focal: we use the recently proposed focal loss [35] as another baseline. (6) SGD schedule: by SGD, we refer to the standard schedule where the learning rates are decayed a constant factor at certain steps; we use a standard learning rate decay schedule.

**Our proposed algorithm and variants.** We test combinations of the following techniques proposed by us. (1) DRW and DRS: following the proposed training Algorithm 1, we use the standard ERM optimization schedule until the last learning rate decay, and then apply re-weighting or re-sampling for optimization in the second stage. (2) LDAM: the proposed Label-Distribution-Aware Margin losses as described in Section 3.2.

When two of these methods can be combined, we will concatenate the acronyms with a dash in between as an abbreviation. The main algorithm we propose is LDAM-DRW. Please refer to Section B for additional implementation details.

## 4.1 Experimental results on IMDB review dataset

IMDB review dataset consists of 50,000 movie reviews for binary sentiment classification [41]. The original dataset contains an evenly distributed number of positive and negative reviews. We manually created an imbalanced training set by removing 90% of negative reviews. We train a two-layer bidirectional LSTM with Adam optimizer [26]. The results are reported in Table 1.

## 4.2 Experimental results on CIFAR

**Imbalanced CIFAR-10 and CIFAR-100.** The original version of CIFAR-10 and CIFAR-100 contains 50,000 training images and 10,000 validation images of size $32 \times 32$ with 10 and 100 classes, respectively. To create their imbalanced version, we reduce the number of training examples per class and keep the validation set unchanged. To ensure that our methods apply to a variety of settings, we consider two types of imbalance: long-tailed imbalance [10] and step imbalance [5]. We use imbalance ratio $\rho$ to denote the ratio between sample sizes of the most frequent and least frequent class, i.e., $\rho = \max_i\{n_i\} / \min_i\{n_i\}$. Long-tailed imbalance follows an exponential decay in sample sizes across different classes. For step imbalance setting, all minority classes have the same sample size, as do all frequent classes. This gives a clear distinction between minority classes and frequent classes, which is particularly useful for ablation study. We further define the fraction of minority classes as $\mu$. By default we set $\mu = 0.5$ for all experiments.

Table 2: Top-1 validation errors of ResNet-32 on imbalanced CIFAR-10 and CIFAR-100. The combination of our two techniques, LDAM-DRW, achieves the best performance, and each of them individually are beneficial when combined with other losses or schedules.

| Dataset | Imbalanced CIFAR-10 | | | | Imbalanced CIFAR-100 | | | |
|---|---|---|---|---|---|---|---|---|
| Imbalance Type | long-tailed | | step | | long-tailed | | step | |
| Imbalance Ratio | 100 | 10 | 100 | 10 | 100 | 10 | 100 | 10 |
| ERM | 29.64 | 13.61 | 36.70 | 17.50 | 61.68 | 44.30 | 61.45 | 45.37 |
| Focal [35] | 29.62 | 13.34 | 36.09 | 16.36 | 61.59 | 44.22 | 61.43 | 46.54 |
| LDAM | 26.65 | 13.04 | 33.42 | 15.00 | 60.40 | 43.09 | 60.42 | 43.73 |
| CB RS | 29.45 | 13.21 | 38.14 | 15.41 | 66.56 | 44.94 | 66.23 | 46.92 |
| CB RW [10] | 27.63 | 13.46 | 38.06 | 16.20 | 66.01 | 42.88 | 78.69 | 47.52 |
| CB Focal [10] | 25.43 | 12.90 | 39.73 | 16.54 | 63.98 | 42.01 | 80.24 | 49.98 |
| HG-DRS | 27.16 | 14.03 | 29.93 | 14.85 | - | - | - | - |
| LDAM-HG-DRS | 24.42 | 12.72 | 24.53 | 12.82 | - | - | - | - |
| M-DRW | 24.94 | 13.57 | 27.67 | 13.17 | 59.49 | 43.78 | 58.91 | 44.72 |
| **LDAM-DRW** | **22.97** | **11.84** | **23.08** | **12.19** | **57.96** | **41.29** | **54.64** | **40.54** |

Table 3: Validation errors on iNaturalist 2018 of various approaches. Our proposed method LDAM-DRW demonstrates significant improvements over the previous state-of-the-arts. We include ERM-DRW and LDAM-SGD for the ablation study.

| Loss | Schedule | Top-1 | Top-5 |
|---|---|---|---|
| ERM | SGD | 42.86 | 21.31 |
| CB Focal [10] | SGD | 38.88 | 18.97 |
| ERM | DRW | 36.27 | 16.55 |
| LDAM | SGD | 35.42 | 16.48 |
| **LDAM** | **DRW** | **32.00** | **14.82** |

We report the top-1 validation error of various methods for imbalanced versions of CIFAR-10 and CIFAR-100 in Table 2. Our proposed approach is LDAM-DRW, but we also include a various combination of our two techniques with other losses and training schedule for our ablation study.

We first show that the proposed label-distribution-aware margin cross-entropy loss is superior to pure cross-entropy loss and one of its variants tailored for imbalanced data, focal loss, while no data-rebalance learning schedule is applied. We also demonstrate that our full pipeline outperforms the previous state-of-the-arts by a large margin. To further demonstrate that the proposed LDAM loss is essential, we compare it with regularizing by a uniform margin across all classes under the setting of cross-entropy loss and hinge loss. We use M-DRW to denote the algorithm that uses a cross-entropy loss with uniform margin [53] to replace LDAM, namely, the $\Delta_j$ in equation (13) is chosen to be a tuned constant that does not depend on the class $j$. Hinge loss (HG) suffers from optimization issues with 100 classes so we constrain its experiment setting with CIFAR-10 only.

**Imbalanced but known test label distribution:** We also test the performance of an extension of our algorithm in the setting where the test label distribution is known but not uniform. Please see Section C.5 for details.

### 4.3 Visual recognition on iNaturalist 2018 and imbalanced Tiny ImageNet

We further verify the effectiveness of our method on large-scale imbalanced datasets. The iNatualist species classification and detection dataset [52] is a real-world large-scale imbalanced dataset which has 437,513 training images with a total of 8,142 classes in its 2018 version. We adopt the official training and validation splits for our experiments. The training datasets have a long-tailed label distribution and the validation set is designed to have a balanced label distribution. We use ResNet-50 as the backbone network across all experiments for iNaturalist 2018. Table 3 summarizes top-1 validation error for iNaturalist 2018. Notably, our full pipeline is able to outperform the ERM baseline

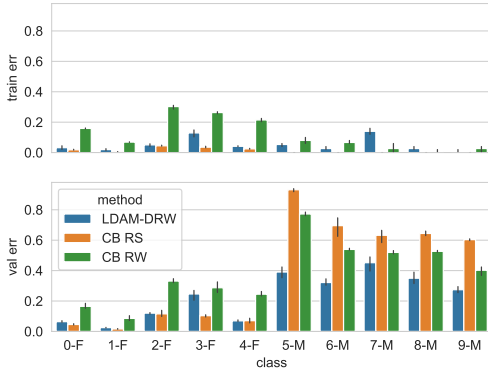

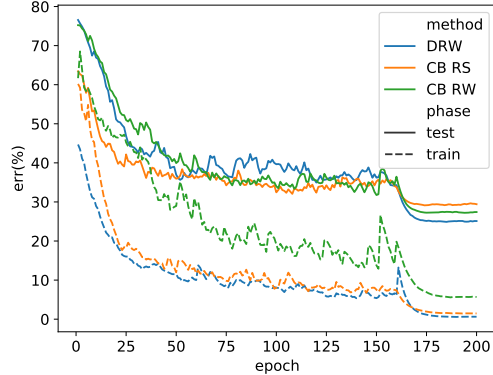

Figure 2: Per-class top-1 error on CIFAR-10 with step imbalance ($\rho = 100, \mu = 0.5$). Classes 0-F to 4-F are frequent classes, and the rest are minority classes. Under this extremely imbalanced setting RW suffers from under-fitting, while RS over-fits on minority examples. On the contrary, the proposed algorithm exhibits great generalization on minority classes while keeping the performance on frequent classes almost unaffected. This suggests we succeeded in regularizing minority classes more strongly.

Figure 3: Imbalanced training errors (dotted lines) and *balanced* test errors (solid lines) on CIFAR-10 with long-tailed imbalance ($\rho = 100$). We anneal decay the learning rate at epoch 160 for all algorithms. Our DRW schedule uses ERM before annealing the learning rate and thus performs worse than RW and RS before that point, as expected. However, it outperforms the others significantly after annealing the learning rate. See Section 4.4 for more analysis.

by 10.86% and previous state-of-the-art by 6.88% in top-1 error. Please refer to Appendix C.2 for results on imbalanced Tiny ImageNet.

### 4.4 Ablation study

**Evaluating generalization on minority classes.** To better understand the improvement of our algorithms, we show per-class errors of different methods in Figure 2 on imbalanced CIFAR-10. Please see the caption there for discussions.

**Evaluating deferred re-balancing schedule.** We compare the learning curves of deferred re-balancing schedule with other baselines in Figure 3. In Figure 6 of Section C.3, we further show that even though ERM in the first stage has slightly worse or comparable balanced test error compared to RW and RS, in fact the features (the last-but-one layer activations) learned by ERM are better than those by RW and RS. This agrees with our intuition that the second stage of DRW, starting from better features, adjusts the decision boundary and locally fine-tunes the features.

## 5 Conclusion

We propose two methods for training on imbalanced datasets, label-distribution-aware margin loss (LDAM), and a deferred re-weighting (DRW) training schedule. Our methods achieve significantly improved performance on a variety of benchmark vision tasks. Furthermore, we provide a theoretically-principled justification of LDAM by showing that it optimizes a uniform-label generalization error bound. For DRW, we believe that deferring re-weighting lets the model avoid the drawbacks associated with re-weighting or re-sampling until after it learns a good initial representation (see some analysis in Figure 3 and Figure 6). However, the precise explanation for DRW's success is not fully theoretically clear, and we leave this as a direction for future work.

**Acknowledgements** Toyota Research Institute ("TRI") provided funds and computational resources to assist the authors with their research but this article solely reflects the opinions and conclusions of its authors and not TRI or any other Toyota entity. We thank Percy Liang and Michael Xie for helpful discussions in various stages of this work.

## Footnotes

[1]Code available at https://github.com/kaidic/LDAM-DRW.

[2]The same technique can also be used for other test label distribution as long as the test label distribution is known. See Section C.5 for some experimental results.

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
