[Supplementary Material · suppfinal_907.pdf]

# A   Missing Proofs and Derivations in Section 3.1

Let $L_{\gamma,j}$ denote the hard margin loss on examples from class $j$:

$$L_{\gamma,j}[f] = \Pr_{x \sim \mathcal{P}_j}[\max_{j' \neq j} f(x)_{j'} > f(x)_j - \gamma]$$

and let $\hat{L}_{\gamma,j}$ denote its empirical variant. For a hypothesis class $\mathcal{F}$, let $\hat{\mathfrak{R}}_j(\mathcal{F})$ denote the empirical Rademacher complexity of its class $j$ margin:

$$\hat{\mathfrak{R}}_j(\mathcal{F}) = \frac{1}{n_j}\mathbb{E}_\sigma\left[\sup_{f \in \mathcal{F}} \sum_{i \in S_j} \sigma_i[f(x_i)_j - \max_{j' \neq j} f(x_i)_{j'}]\right]$$

where $\sigma$ is a vector of i.i.d. uniform $\{-1, +1\}$ bits. The following formal versiom of Theorem 1 bounds the balanced-class generalization $\mathcal{P}_{\text{bal}}$ using samples from $\mathcal{P}$.

**Theorem 2.** *With probability $1 - \delta$ over the randomness of the training data, for all choices of class-dependent margins $\gamma_1, \ldots, \gamma_k > 0$, all hypotheses $f \in \mathcal{F}$ will have balanced-class generalization bounded by*

$$L_{\text{bal}}[f] \leq \frac{1}{k}\left(\sum_{j=1}^k \hat{L}_{\gamma_j,j}[f] + \frac{4}{\gamma_j}\hat{\mathfrak{R}}_j(\mathcal{F}) + \epsilon_j(\gamma_j)\right)$$

*where $\epsilon_j(\gamma) \triangleq \sqrt{\frac{\log\log_2(\frac{2\max_{x \in \mathcal{X}, f \in \mathcal{F}} |f(x)|}{\gamma}) + \log\frac{2c}{\delta}}{n_j}}$ is typically a low-order term in $n_j$. Concretely, the Rademacher complexity $\hat{\mathfrak{R}}_j(\mathcal{F})$ will typically scale as $\sqrt{\frac{C(\mathcal{F})}{n_j}}$ for some complexity measure $C(\mathcal{F})$, in which case*

$$L_{\text{bal}}[f] \leq \frac{1}{k}\left(\sum_{j=1}^k \hat{L}_{\gamma_j,j}[f] + \frac{4}{\gamma_j}\sqrt{\frac{C(\mathcal{F})}{n_j}} + \epsilon_j(\gamma_j)\right)$$

*Proof.* We will prove generalization separately for each class $j$ and then union bound over all classes.

Let $L_j[f]$ denote the test $0 - 1$ error of classifier $f$ on examples drawn from $\mathcal{P}_j$. As the examples for class $j$ is a set of $n_j$ i.i.d. draws from the conditional distribution $\mathcal{P}_j$, we can apply the standard margin-based generalization bound (Theorem 2 of [24]) to obtain with probability $1 - \delta/k$, for all choices of $\gamma_j > 0$ and $f \in \mathcal{F}$,

$$L_j[f] \leq \hat{L}_{\gamma_j,j} + \frac{4}{\gamma_j}\hat{\mathfrak{R}}_j(\mathcal{F}) + \sqrt{\frac{\log\log_2(\frac{2\max_{x \in \mathcal{X}, f \in \mathcal{F}} |f(x)|}{\gamma_j})}{n_j}} + \sqrt{\frac{\log\frac{2c}{\delta}}{n_j}} \quad (14)$$

Now since $L_{\text{bal}} = \frac{1}{k}\sum_{j=1}^k L_j$, we can union bound over all classes and average (14) to get the desired result. $\qquad\square$

We will now show that in the case of $k = 2$ classes, it is always possible to shift the margins in order to optimize the generalization bound of Theorem 2 by adding bias terms.

**Theorem 3.** *For binary classification, let $\mathcal{F}$ be a hypothesis class of neural networks with a bias term, i.e. $\mathcal{F} = \{f + b\}$ where $f$ is a neural net function and $b \in \mathbb{R}^2$ is a bias, with Rademacher complexity upper bound $\hat{\mathfrak{R}}_j(\mathcal{F}) \leq \sqrt{\frac{C(\mathcal{F})}{n_j}}$. Suppose some classifier $f \in \mathcal{F}$ can achieve a total sum of margins $\gamma_1' + \gamma_2' = \beta$ with $\gamma_1', \gamma_2' > 0$. Then there exists a classifier $f^\star \in \mathcal{F}$ with margins*

$$\gamma_1^\star = \frac{\beta n_2^{1/4}}{n_1^{1/4} + n_2^{1/4}} \;,\; \gamma_2^\star = \frac{\beta n_1^{1/4}}{n_1^{1/4} + n_2^{1/4}}$$

*which with probability $1 - \delta$ obtains the optimal generalization guarantees for Theorem 2:*

$$L_{\text{bal}}[f^\star] \leq \min_{\gamma_1 + \gamma_2 = \beta}\left(\frac{2}{\gamma_1}\sqrt{\frac{C(\mathcal{F})}{n_1}} + \frac{2}{\gamma_2}\sqrt{\frac{C(\mathcal{F})}{n_2}}\right) + \epsilon(\gamma_1^\star) + \epsilon(\gamma_2^\star)$$

*where $\epsilon$ is defined in Theorem 2. Furthermore, this $f^\star$ is obtained via $f + b^\star$ for some bias $b^\star$.*

|         | (a) $\rho = 10$ | (b) $\rho = 100$ | (c) $\rho = 10, \ \mu = 0.5$ |
|---------|-----------------|------------------|------------------------------|

Figure 4: Number of training examples per class in artificially created imbalanced CIFAR-10 datasets. Fig. 4a and Fig. 4b belong to long-tailed imbalance type and Fig. 4c is a step imbalance distribution.

*Proof.* For our bias $b^\star$, we simply choose $b_1^\star = (\gamma_1^\star - \gamma_1')/2$, $b_2^\star = -(\gamma_1^\star - \gamma_1')/2$. Now note that adding a bias term simply shifts the margins for class 1 by $b_1^\star - b_2^\star$, giving a new margin of $\gamma_2^\star$. Likewise, the margin for class 2 becomes

$$b_2^\star - b_1^\star + \gamma_2' = \gamma_2' - \gamma_1^\star + \gamma_1' = \beta - \gamma_1^\star = \gamma_2^\star$$

Now we apply Theorem 2 to get with probability $1 - \delta$ the generalization error bound

$$L_{\text{bal}}[f^\star] \le \frac{2}{\gamma_1^\star}\sqrt{\frac{\mathrm{C}(\mathcal{F})}{n_1}} + \frac{2}{\gamma_2^\star}\sqrt{\frac{\mathrm{C}(\mathcal{F})}{n_2}} + \epsilon(\gamma_1^\star) + \epsilon(\gamma_2^\star)$$

To see that $\gamma_1^\star, \gamma_2^\star$ indeed solve

$$\min_{\gamma_1 + \gamma_2 = \beta} \frac{1}{\gamma_1}\sqrt{\frac{1}{n_1}} + \frac{1}{\gamma_2}\sqrt{\frac{1}{n_2}}$$

we can substitute $\gamma_2 = \beta - \gamma_1$ into the expression and set the derivative to 0, obtaining

$$\frac{1}{(\beta - \gamma_1)^2\sqrt{n_2}} - \frac{1}{\gamma_1^2\sqrt{n_1}} = 0$$

Solving gives $\gamma_1^\star$. $\qquad\qquad\square$

# B   Implementation details

**Label distributions.** Some example distributions of our artificially created imbalance are shown in Figure 4.

**Implementation details for CIFAR.** For CIFAR-10 and CIFAR-100, we follow the simple data augmentation in [19] for training: 4 pixels are padded on each side, and a $32 \times 32$ crop is randomly sampled from the padded image or its horizontal flip. We use ResNet-32 [19] as our base network, and use stochastic gradient descend with momentum of 0.9, weight decay of $2 \times 10^{-4}$ for training. The model is trained with a batch size of 128 for 200 epochs. For fair comparison, we use an initial learning rate of 0.1, then decay by 0.01 at the 160th epoch and again at the 180th epoch. We also use linear warm-up learning rate schedule [14] for the first 5 epochs for fair comparison. Notice that the warm-up trick is essential for the training of re-weighting, but it won't affect other algorithms in our experiments. We tune $C$ to normalize $\Delta_j$ so that the largest enforced margin is 0.5.

**Implementation details for Tiny ImageNet.** For Tiny ImageNet, we perform simple horizontal flips, taking random crops of size $64 \times 64$ from images padded by 8 pixels on each side. We perform 1 crop test with the validation images. We use ResNet-18 [19] as our base network, and use stochastic gradient descend with momentum of 0.9, weight decay of $2 \times 10^{-4}$ for training. We train the model using a batch size of 128 for 120 epochs with a initial learning rate of 0.1. We decay the learning rate by 0.1 at epoch 90. We tune $C$ to normalize $\Delta_j$ so that the largest enforced margin is 0.5.

**Implementation details for iNaturalist 2018.** On iNaturalist 2018, we followed the same training strategy used by [19] and trained ResNet-50 with 4 Tesla V100 GPUs. Each image is first resized by setting the shorter side to 256 pixels, and then a $224 \times 224$ crop is randomly sampled from an

Figure 5: Visualization of feature distribution of different methods. We constrain the feature dimension to be three and normalize it for better illustration. The top row has the feature distribution on the training set and the second row the feature distributions on the validation set. We can see that LDAM appears to have more separate training features compared to the other methods. We note this visualization is only supposed to provide qualitative intuitions, and the differences between our methods and other methods may be more significant for harder tasks with higher feature dimension. (For example, here the accuracies of re-weighting and LDAM are very similar, whereas for large-scale datasets with higher feature dimensions, the gap is significantly larger.)

Table 4: Validation error on imbalanced Tiny ImageNet with different loss functions and training schedules.

| Imbalance Type | | long-tailed | | | | step | | | |
|---|---|---|---|---|---|---|---|---|---|
| Imbalance Ratio | | 100 | | 10 | | 100 | | 10 | |
| Loss | Schedule | Top-1 | Top-5 | Top-1 | Top-5 | Top-1 | Top-5 | Top-1 | Top-5 |
| ERM | SGD | 66.19 | 42.63 | 50.33 | 26.68 | 63.82 | 44.09 | 50.89 | 27.06 |
| CB SM | SGD | 72.72 | 52.62 | 51.58 | 28.91 | 74.90 | 59.14 | 54.51 | 33.23 |
| ERM | DRW | 64.57 | 40.79 | 50.03 | 26.19 | 62.36 | 40.84 | 49.17 | 25.91 |
| LDAM | SGD | 64.04 | 40.46 | 48.08 | 24.80 | 62.54 | 39.27 | 49.08 | 24.52 |
| LDAM | DRW | **62.53** | **39.06** | **47.22** | **23.84** | **60.63** | **38.12** | **47.43** | **23.26** |

image or its horizontal flip. We train the network for 90 epochs with an initial learning rate of 0.1. We anneal the learning rate at epoch 30 and 60. For our two-stage training schedule, we rebalance the training data starting from epoch 60. We tune $C$ to normalize $\Delta_j$ so that the largest enforced margin is 0.3.

## C  Additional Results

### C.1  Feature visualization

To have a better understanding of our proposed LDAM loss, we use a toy example to visualize feature distributions trained under different schemes. We train a 7-layer CNN as adopted in [39] on MNIST [30] with step imbalance setting ($\rho = 100, \mu = 0.5$). For a more intuitive visualization, we constrain the feature dimension to 3 and normalize the feature before feeding it into the final fully-connected layer, allowing us to scatter the features on a unit hyper-sphere in a 3D frame. The visualization is shown in Figure 5 with additional discussion in the caption.

Figure 6: In the setting of training mbalanced CIFAR-10 dataset with step imbalance of $\rho = 100, \mu = 0.5$, to test the quality of the features obtained by the ERM, RW and RS before annealing the learning rate, we use a subset of the *balanced* validation dataset to train linear classifiers on top of the features, and evaluate the per-class validation error on the rest of the validation data. (Little over-fitting in training the linear classifier is observed.) The left-5 classes are frequent and denoted with -F. The features obtained from ERM setting has the strongest performance, confirming our intuition that the second stage of DRW starts from better features. In the second stage, DRW re-weights the example again, adjusting the decision boundary and locally fine-tuning the features.

## C.2 Visual Recognition on imbalanced Tiny ImageNet

In addition to artificial imbalanced CIFAR, we further verify the effectiveness of our method on artificial imbalanced Tiny ImageNet. The Tiny ImageNet dataset has 200 classes. Each class has 500 training images and 50 validation images of size $64 \times 64$. We use the same strategy described above to create long-tailed and step imbalance versions of Tiny ImageNet. The results are presented in Table 4. While Class-Balanced Softmax performs worse than the ERM baseline, the proposed LDAM and DRW demonstrate consistent improvements over ERM.

## C.3 Comparing feature extractors trained by different schemes

As discussed in Section 4.4, we train a linear classifier on features extracted by backbone filters pretrained under different schemes. We could conclude that for highly imbalanced settings (step imbalance with $\rho = 100, \mu = 0.5$), backbone networks trained by ERM learns the most expressive feature embedding compared with the other two methods, as shown in Figure 6.

## C.4 Comparing DRW and DRS

Our proposed deferred re-balancing optimization schedule can be combined with either re-weighting or re-sampling. We use re-weighting as the default choice in the main paper. Here we demonstrate through Table 5 that re-weighting and re-sampling exhibit similar performance when combined with deferred re-balancing scheme. This result could be explained by the fact that the second stage does not move the weights far. Re-balancing in the second stage mostly re-adjusts the decision boundary and thus there is no significant difference between using re-weighting or re-sampling for the second stage.

## C.5 Imbalanced Test Label Distributions

Though the majority of our experiments follow the uniform test distribution setting, it could be extended to imbalanced test distribution naturally. Suppose the number of training examples in class $i$ is denoted by $n_i$ and the number of test examples in class $i$ is denoted by $n'_i$, then we could adapt

Table 5: Top-1 validation error of ResNet-32 trained with different training schedules on imbalanced CIFAR-10 and CIFAR-100.

| Dataset Name | Imbalanced CIFAR-10 | | | | Imbalanced CIFAR-100 | | | |
|---|---|---|---|---|---|---|---|---|
| Imbalance Type | long-tailed | | step | | long-tailed | | step | |
| Imbalance Ratio | 100 | 10 | 100 | 10 | 100 | 10 | 100 | 10 |
| ERM | 29.64 | 13.61 | 36.70 | 17.50 | 61.68 | 44.30 | 61.05 | 45.37 |
| DRW | 25.14 | 13.12 | 28.40 | 14.49 | 59.34 | 42.68 | 58.86 | 42.78 |
| DRS | 25.50 | 13.28 | 27.97 | 14.83 | 59.67 | 42.74 | 58.65 | 43.21 |

(a) train set distribution     (b) val set distribution 1     (c) val set distribution 2

Figure 7: Example distributions when train and test distributions are both imbalanced. As discussed in C.5 we run two random seeds for generating test distributions. Here Figure 7b denotes the left column in Table 6.

the LDAM simply by encouraging the margin $\Delta_i$ for class $i$ with

$$\Delta_j \propto \left( \frac{n_i'}{n_i} \right)^{1/4} \tag{15}$$

To complement our main result, In Table 6, we demonstrate that this extended algorithm can also work well when the test distribution is imbalanced. We use the same rule as described in Section 4 to generate imbalanced test label distribution and then permute randomly the frequency of the labels (so that the training label distribution is very different from the test label distribution.). For example, in the experiment shown in Figure 6, the training label distribution of the column of "long-tailed with $\rho = 100$" follows Figure 7a (which is the same as Figure 4b) whereas the test label distribution is shown in Figure 7b and Figure 7c. For each of the settings reported in Table 6, we have run it with two different random seeds for generating the test label distribution, and we see qualitatively similar results. We refer to our code for the precise label distribution generated in the experiments.[3]

Table 6: Top-1 validation error of ResNet-32 on imbalanced training and imbalanced validation scheme for CIFAR-10. See Section C.5 for details.

| Imbalance Type | long-tailed | | | | step | | | |
|---|---|---|---|---|---|---|---|---|
| Imbalance Ratio Train | 100 | | 10 | | 100 | | 10 | |
| Imbalance Ratio Val | 100 | 100 | 10 | 10 | 100 | 100 | 10 | 10 |
| ERM | 30.99 | 28.45 | 13.08 | 13.12 | 24.55 | 28.63 | 10.34 | 11.67 |
| CB-RW | 20.86 | 26.19 | 10.70 | 11.93 | 35.76 | 31.35 | 9.82 | 11.02 |
| LDAM-DRW | **14.40** | **12.95** | **10.12** | **10.62** | **10.30** | **9.54** | **7.51** | **7.82** |

## Footnotes

[3]Code available at `https://github.com/kaidic/LDAM-DRW`.