[Reviews · NeurIPS 2019]

Reviewer 1



This is an interesting paper with solid and novel work. 1. Novelty. The authors start from label distribution independent margin theory to distribution aware margin generalization bound, from which the optimal distribution aware margin is derived. Inspired by the proposed label distribution aware margin theory, the authors propose the LDAM loss to adjust the margin between frequent and minority classes. The class-distribution-aware margin trade-off theory is profound and novel. 2. Quality. The main mathematical tool applied is margin theory. The mathematical statement is simplified, like Eq. (3), however, just enough to illustrate. I didn't check the proof carefully provided by the authors. 3. Clarity. The paper is well-written and organized. 4. Significance. The proposed method is simple yet effective. Inspired from margin theory the proposed method is theoretically sound. Also the authors conduct carefully designed experiments, which validates the feasibility.

Reviewer 2



This is a new solution to an old problem. The problem is somewhat significant. The paper's goal is clear and is overall well written. The theoretical analysis is sound and intuitive, though I did not check the proofs. The simulation study, including an Ablation study, is fairly thorough. However, I find some details are missing. 1. It is unclear to me why the loss function (10) enforces the desired margin in (9). This seems to be a missing piece of puzzle. Some better and intuitive explanation (perhaps a direct calculation) is needed. 2. I wonder what exactly is showing in Figure 2. What is "feature distribution of different methods"? Isn't that the output X_i*b where b is the coefficient matrix? 3. I am puzzled by some of the discussion in 3.3. On the one hand, the authors proposed "Deferred Re-balancing Optimization Schedule", which is to " first trains using vanilla ERM with the LDAM loss before annealing the learning rate, and then deploys a re-weighted LDAM loss with a smaller learning rate." However, they also mentioned that " the second stage does not move the weights very far." If the second stage does not move the weight by much, then shouldn't the vanilla ERM with LDAM loss work well enough? I don't think the results have any issue, but the way the motivation of this new strategy is presented needs a fix up. Edit: I have read the author response and changed my score from 6 to 7.

Reviewer 3



+ LDAM aims to put regularization on the margins (i.e. the minimum distance of data samples to the decision boundary) of minority classes in order to improve the generalizability of the model towards minority classes during the test time, in which the value of margin is set to be proportional to the number of samples for each class thus the LDAM is label-distribution-aware. DRW runs reweighting and LDAM with smaller learning rate in order to perform fine-tuning on the model after an initial stage of training. Although without any theoretical justification, its efficacy is successfully proven across various experiments. These ideas are novel and shown to provide better superior performance, even avoid overfitting for frequent classes, in comparison to naive re-weighting/re-sampling techniques and other baselines (e.g. Focal loss, CB [8]). - There is no description on how to decide the hyperparameter C (which is used in equation.13 for adjusting the class-dependent margins). It is also required to have an analysis on the sensitivity of performance with respect to C. Additionally, as in both stages of Algorithm.1 LDAM is used, should there be different values of C? - How is the LDAM-HG-DRS in Table.1 implemented? - While CB [8] is the main baseline used in this paper for comparison, it calls for a clearer explanation on the difference between CB [8] and the proposed method. To be detailed, although CB [8] is a concurrent work (published in CVPR-19) to this submission and it models the class-imbalance problem from a slightly different perspective, when we examine the equation.12 & equation.13 in this submission and have comparison w.r.t. CB+softmax, they seem to be quite similar thus here requires more insight to point out why and how the proposed LDAM brings superior benefits to the performance. - As we can see from Table.2, it seems that the main boost of performance is stemmed from the DRW (deferred re-weighting) mechanism, together with the similarity between CB [8] and the proposed LDAM, we would need an additional baseline, i.e. CB+DRW, to clarify the contribution of LDAM. ** after rebuttal ** My review comments and concerns are well addressed by the authors. Therefore I would stick to my positive opinion on this submission. However, I agree with other reviewers that currently the derivations of Equation. 5, 6, 7, 8 are simply based on the binary classification, there should be more justification on the case of multi-class. I change my rating from 7 to 6 accordingly.

Reviewer 4



The key point of this paper is the equation (10) which is inspired by the trade-off between the class margins for BINARY classification. The loss function, LDAM takes the class margins into account so that the minority classes will enjoy a larger margin. The best numerical results come from LDAM-DRW scheme. I can't not find a very convincing argument for deferred re-weight scheme. The numerical result outperform than current popular strategies for dealing with imbalanced dataset. However, it is only tested on the computer vision learning task.

[Author Response · NeurIPS 2019]

We thank the reviewers for their insightful comments and constructive feedback. As the reviewers mentioned, our work
shows the following strengths. (1) The theoretical analysis is "profound and novel" [R3,R4,R5]. (2) Experiments are
designed "thoroughly" and "carefully" which "verifies the feasibility" [R3,R4]. (3) The paper is "well-written and
organized" [R3,R4]. We will answer the major points below and address all remaining ones in the final version.

**[R3]:** "For eq. (3) (4) (5), the first item on the right-hand side, $\sqrt{\frac{C(F)}{n_j}}$ or $\frac{C(F)}{\sqrt{n_j}}$?"
• This depends on how $C(F)$ is defined. If it is defined to be the Rademacher complexity, then the former is correct.

**[R3]:** I suggest the authors to polish up the Figure 1.
• Thanks for the suggestion! We'll update with a better one for the final version.

**[R3]:** ""Hinge loss (HG) does not work well with 100 classes", what you mean by not work well?"
• When trained on CIFAR-100, Hinge loss seems to suffer from optimization issues — the training accuracy is at most
about 80%. Thus we didn't report the test accuracy because the failure here is of a different nature.

**[R4,R6]:** "It is unclear to me why the loss function (10) enforces the desired margin in (9)."; "Provide a strong
justification for the equation (10)"
• The Hinge loss in (10) achieves its minimum value zero only if the margin is at least $\Delta_y$. Recall that the margin is
defined to be $\gamma = z_y - \max_{j \neq y} z_j$. Therefore, Hinge loss $= \max\{\Delta_y - \gamma, 0\} = 0$ if and only if $\gamma \geq \Delta_j$. Hinge loss is
a standard loss that encourages margins in the context of SVM.[1] We extend it to allow label-dependent margins.

**[R4]:** "I wonder what exactly is showing in Figure 2."
• We visualize the distributions of the last-but-one layer of the neural network, which are referred to as the features.
Please refer to the details in L230-L235. We will clarify more in the final version.

**[R4,R6]:** "If the second stage does not move the weight by much, shouldn't the ERM with LDAM loss work well
enough?"; "Provide a better why DRW is important?"
• We believe that the second stage with smaller learning rate serves as a fine-tuning-like process to capture sophisticated
details in each class. Thus in the second stage, emphasizing rare examples are important, because without it, the training
accuracies for all the classes can not be approximately 100%. (Relatively smaller movements in the second stage could
also change the performance by more than a few percents.) With the initial large learning rate in the first stage, by
contrast, the network learns the shared patterns/features shared across all tasks, and therefore it would be better to train
with all the examples with uniform weights. Such phenomenon/intuitions were also observed[2] and justified in recent
works[3]. We realized this from the ablation study in Fig. 6 in Appendix, which shows that the features learned in the
first stage with ERM are better than those with re-weighting.

**[R5]:** "How to decide the hyperparameter C? How is the LDAM-HG-DRS in Table.1 implemented?"
• We tune $C$ as a hyper-parameter for each dataset. In particular, we use $C = 0.5$ for all CIFAR-10 and CIFAR-100
experiments, and $C = 0.3$ for all iNaturalist experiments. Regarding the LDAM-HG-DRS implementation, we follow
Eq. (10) to implement Hinge loss. Here DRS means the delayed re-sampling strategy.

**[R5]:** "CB+Softmax and LDAM seem to be quite similar"; "it seems that the main boost of performance is stemmed
from the DRW (deferred re-weighting)"; additional baseline CB+DRW.
• We'd like first to clarify that CB only re-weights the losses, and therefore is a re-weighting scheme more similar to
vanilla re-weighting than to LDAM (which is a new loss). DRW, a deferred re-weighting scheme that we proposed, is
an improved version of CB or vanilla re-weighting, and is orthogonal to LDAM. In Tab. 2, we see that either using
LDAM alone (4th row), or DRW alone (3rd row), on top of the ERM baseline, can outperform prior work. LDAM alone
(3.5% improvement) is slightly more useful than DRW alone (2.6%), and together, they give 6.8% improvement. Thus
we don't agree that the main boost stems from DRW. We found CB+DRW does not outperform DRW alone, which also
suggests that DRW is a better re-weighting scheme.

**[R6]:** Test the proposed method for more general machine learning tasks.
• Thank you for your suggestion. We selected these datasets (1) to compare with related works, (2) because they are
challenging, (3) because they are representative of ubiquitous real-world dataset imbalance issues. Nonetheless, we add
one additional sentiment analysis experiment on the Large Movie Review (IMDB) Dataset, a popular and standard
task in NLP. We manually created an imbalanced training set by removing 90% of negative reviews. We train a 2-layer
bidirectional LSTM with Adam optimizer. Test accuracy of different methods are listed as follows: ERM: 63.18,
Re-weight: 76.34, Re-sample: 73.50, LDAM: 82.16. Thus our conclusions hold on other tasks. We will add this result
to the final version of the paper.

## Footnotes

[1] Wikipedia contributors. "Hinge loss." Wikipedia, The Free Encyclopedia.

[2] Nakkiran, Preetum, et al. "SGD on Neural Networks Learns Functions of Increasing Complexity."

[3] Li, Yuanzhi, et al. "Towards Explaining the Regularization Effect of Initial Large Learning Rate in Training Neural Networks."


[Meta-Review · NeurIPS 2019]

The authors propose to solve imbalanced classification problems by requiring uneven margins for each class of examples. The reviewers agree that the proposed view is novel and comes with some insights from the theoretical side. The authors then design a new loss to achieve the uneven margins and apply the loss within a two-stage algorithm to achieve promising performance on some image data sets. The promising performance on the important problem of imbalanced classification makes the paper sufficiently interesting for the NeurIPS audience. The authors are somehow encouraged to clarify the gap between the theoretical results on binary classification and the algorithmic results on multi-class classification.